# Hippocampal and Cerebellar Changes in Acute Restraint Stress and the Impact of Pretreatment with Ceftriaxone

**DOI:** 10.3390/brainsci10040193

**Published:** 2020-03-25

**Authors:** Shaimaa N. Amin, Sherif S. Hassan, Ahmed S. Khashaba, Magdy F. Youakim, Noha S. Abdel Latif, Laila A. Rashed, Hanan D. Yassa

**Affiliations:** 1Department of Anatomy, Physiology and Biochemistry, Faculty of Medicine, Hashemite University, Zarqa 13133, Jordan; 2Department of Medical Physiology, Faculty of Medicine, Cairo University, Cairo 11451, Egypt; 3Department of Medical Education, School of Medicine, California University of Science & Medicine, San Bernardino, CA 82408, USA; 4Department of Anatomy, Faculty of Medicine, Cairo University, Cairo 11451, Egypt; drmagdy@hotmail.com; 5Department of Basic Sciences, Riyadh Elm University, Riyadh 12734, Saudi Arabia; ahmedkhashaba@riyadh.edu.sa; 6Department of Medical Pharmacology, Faculty of Medicine, Cairo University, Cairo 11451, Egypt; noha.gomaa@kasralainy.edu.eg; 7Department of Biochemistry, Faculty of Medicine, Cairo University, Cairo 11451, Egypt; lailarashed@kasralainy.edu.eg; 8Department of Anatomy and Embryology, Faculty of Medicine, Beni-Suef University, Beni Suef 62511, Egypt; hanan_yassa2000@yahoo.com

**Keywords:** acute restraint stress, hippocampus, cerebellum, tissue markers, histopathological examination

## Abstract

Acute restraint stress (ARS) is an unavoidable stress situation and may be encountered in different clinical situations. The aim of the current study was to investigate the effects of ARS on the hippocampus and cerebellum, assess the impact of these effects on the behavior and cognitive function, and determine whether pretreatment with ceftriaxone would attenuate the damages produced by ARS on the hippocampus and cerebellum. Four groups of male mice were included in this study: The control group, ARS group, ceftriaxone group, and ARS + ceftriaxone group. Tail suspension test, Y-maze task, and open field tests were used to assess depression, working spatial memory, and anxiety. The biochemical analyses included measurements of serum cortisol, tumor necrotic factor (TNF), interleukin-6, hippocampal expression of bone morphogenetic protein 9 (BMP9), lysosomal-associated membrane protein 1 (LAMP1), glutamate transporter 1 (GLT1), heat shock protein 90, cerebellar expression of S100 protein, glutamic acid decarboxylase (GAD), and carbon anhydrase. Histopathological examination of the brain sections was conducted on the hippocampus and cerebellum by hematoxylin and eosin stains in addition to ultrastructure evaluation using electron microscopy. Our results suggested that ceftriaxone had neuroprotective properties by attenuating the effects of ARS on the hippocampus and cerebellum in mice. This effect was demonstrated by the improvement in the cognitive and behavioral tests as well as by the preservation of the hippocampal and cerebellar architecture.

## 1. Introduction

Acute restraint strain (ARS) is an inevitable stress situation that might cause autonomic and behavioral alternation [1]. The hippocampus is necessary for the formation of declarative memory in humans, spatial memory in rodents, memory consolidation, reconsolidation, long-term memory persistence, novelty detection, habituation, and long-term potentiation [2]. ARS adversely affects the hippocampus both structurally and functionally [3,4].

The cerebellum is the area of the hind brain that plays an important role in cognitive and emotional processes [5]. The dysmetria of thought theory holds that the cerebellum modulates behavior just as the cerebellum controls the pace, intensity, rhythm, and precision of motions, and the speed, ability, continuity, and appropriateness of mental or cognitive processes [6]. Cerebellar impairment has been shown in various psychological conditions, including post-traumatic stress disorder (PTSD) and anxiety disorders [7].

Ceftriaxone is an antibiotic that has shown neuroprotective effects in animal models of stroke [8], traumatic brain injuries (TBI), Alzheimer’s disease [9], and Parkinson’s disease [10].

The aim of the current study was to investigate the effects of ARS on the hippocampus and cerebellum, assess the impact of these effects on the behavior and cognitive function, and evaluate whether pretreatment with ceftriaxone would attenuate the damages produced by ARS on the hippocampus and cerebellum.

## 2. Materials and Methods

### 2.1. Experimental Animals and Study Design 

In total, 24 male albino mice aged 10 weeks (25–30 g) constituted the animal model of the current study. The guidelines of the Ethical and Scientific Committee of the Department of Medical Physiology at the Faculty of Medicine, Cairo University were followed. Rats were divided into 4 groups with 6 mice per group:

-*Vehicle control group:* Where mice received normal saline (0.9% NaCL solution) intraperitoneally (i.p), with a 24-h interval for 3 consecutive days.

-*Acute restraint stress (ARS) group:* Where each mouse experienced a single stress session following 18 h of fasting (food deprivation). Each stress session consisted of 2.5 h of immobilization by firmly securing all four limbs of each mouse to a grid using a quartz tape [11].

-*Ceftriaxone group:* Received ceftriaxone (Pfizer, NY, USA), i.p., as a single dosage of 200 mg/kg/day dissolved in normal saline [12] for 3 consecutive days.

-*ARS + ceftriaxone group:* Where each mouse experienced ARS (in a similar way to that in the ARS group) and received 3 doses of ceftriaxone (24 h before ARS, 1 h before ARS, and 24 h after ARS).

### 2.2. Cognitive and Behavioral Evaluation

The following tests were performed before drug administration and grouping of mice and after acclimatization to the laboratory environment to avoid previously reported problems in the literature. The tests were then repeated 24 h after exposure to ARS.

*Open field test:* Open field testing measures locomotion, exploration, and anxiety. A wooden box (1 m × 1 m × 0.5 m) was divided by a marker pen into equally spaced squares. A mouse was positioned in the center of the open field and was examined in a quiet room illuminated by controlled light for 5 min [13]. The behavior of the mouse was evaluated for line crossing, center square entries, center square time, rearing, stretch attending postures, grooming, freezing, urination, and defecation.

*Tail suspension test (TST):* It tests depression-like behavior in mice. Mice were held 50 cm above the floor by adhesive tape positioned about 1 cm from the tip of the tail. Immobility time was recorded for a period of 6 min. Mice were considered immobile only when they hung passively and became completely immobile [14].

*Y-maze test:* Mice were placed in the middle of the maze and were allowed to explore the three arms for 8 min to check the working memory of the space. The three successive decisions taken by the mouse with three different arms were counted as a correct choice. The alternation score was calculated by dividing the total number of alternations by the total number of choices minus 2 multiplied by 100 [15]. 

After 24 h, serum samples were received from the retro-orbital sinuses. The mice were euthanized by cervical decapitation. The cerebella and hippocampi were dissected, and their tissues were used for biochemical analyses and histopathological examination.

### 2.3. Biochemical Analyses

Serum was analyzed for cortisol, tumor necrotic factor-alpha (TNF-α), and interleukin-6 (IL-6) and cortisol levels by ELISA (Quantikine R&D system USA) as instructed by the manufacturer. 

Hippocampus tissue was assessed for bone morphogenetic protein 9 (BMP9), lysosomal-associated membrane protein 1 (LAMP1), glutamate transporter 1 (GLT1) and heat shock protein 90 and cerebellar tissue was evaluated for S100 protein levels, GLT1, glutamic acid decarboxylase (GAD) and carbonic anhydrase by RT-PCR (Fisher Scientific, Waltham, Massachusetts, United States).

#### 2.3.1. Quantitative Analysis of Gene Expression by Real-Time PCR

*Total RNA extraction:* Total RNA was extracted from the homogenized tissue using the SV Total RNA Isolation System (Promega, Madison, WI, USA) according to the manufacturer’s instructions. RNA concentrations and purity were calculated using an ultraviolet spectrophotometer (BioTek Instruments, Winooski, VT, USA).

#### 2.3.2. Complementary DNA (cDNA) Synthesis 

The cDNA was synthesized from 1 μg of RNA using the SuperScript III First-Strand Synthesis System according to the manufacturer’s protocol (# K1621, Fermentas, Waltham, MA, USA). To summarize, 1 μg of total RNA was combined with 50 μM oligo (dT) 20, 50 ng/μL random primers, and 10 mM dNTP. The total mixture volume was 10 μL. The mixture was incubated at 56 °C for 5 min, then put in ice for another 3 min. The total mixture comprising 2 μL of 10× RT solution, 4 μL of 25 mM MgCl_2_, 2 μL of 0.1 M DTT, and 1 μL of SuperScript^®^ III RT (200 U/μL) was applied to the total mixture and incubated at 25 °C for 10 min followed by another 50 min at 50 °C.

#### 2.3.3. Real-Time Quantitative PCR

Real-time PCR amplification and analysis were performed using the Applied Biosystem software version 3.1 StepOne (Thermo Fisher Scientific, Waltham, MA, USA). The reaction included SYBR Green Master Mix (Applied Biosystems Foster City, California, United States), a gene-specific primer pair (Table 1), which was designed with Gene Runner Software (Hasting Software, Inc., Hasting, NY, USA) from RNA sequences from a gene bank. All primer units had a calculated annealing temperature of 60 °C. Quantitative RT-PCR was carried out in a 25-μL reaction volume consisting of 2× SYBR Green PCR Master Mix (Applied Biosystems, Foster City, California, United States), 900 nM of each primer, and 2 μL of cDNA. The amplification settings were: 2 min at 50 °C 10 min at 95 °C and 40 cycles of denaturation at 15 s, and annealing/extension at 60 °C for 10 min (Foster City, CA; USA). The relative expression of the examined mRNA gene was determined using the comparative Ct test. Both values were converted to beta-actin, which was used as a control housekeeping gene and identified as a fold change over the background levels observed in the diseased groups.

#### 2.3.4. Detection of GLT1 Protein by the Western Blot Technique (Using aV3 Western Workflow™ Complete System, Bio-Rad^®^ Hercules, CA, USA): 

To summarize, proteins were extracted from tissue homogenates using ice-cold radioimmunoprecipitation assay (RIPA) buffer supplemented with phosphatase and protease inhibitors (50 mmol/L sodium vanadate, 0.5 mM phenylmethylsulphonyl fluoride, 2 mg/mL aprotinin, and 0.5 mg/mL leupeptin), then centrifuged at 12,000 rpm for 20 min. The protein concentration for each sample was determined using Bradford assay. Equal amounts of protein (20–30 µg of total protein) were separated by SDS/polyacrylamide gel electrophoresis (10% acrylamide gel) using a Bio-Rad Mini-Protein II system. The protein was transferred to polyvinylidene difluoride membranes (Pierce, Rockford, IL, USA) with a Bio-Rad Trans-Blot system. After transfer, the membranes were washed with PBS and were blocked for 1 h at room temperature with 5% (w/v) skimmed milk powder in PBS. The manufacturer’s instructions were followed for the primary antibody reactions. Following blocking, the blots were developed using antibodies for GLT1and beta actin supplied by Thermoscientific (Rockford, Illinois, USA) and incubated overnight at pH 7.6 at 4 °C with gentle shaking. After washing, peroxidase-labeled secondary antibodies were added, and the membranes were incubated at 37 °C for 1 h. Band intensity was analyzed by a ChemiDocTM imaging system with Image LabTM software version 5.1 (Bio-Rad Laboratories Inc., Hercules, CA, USA). The results were expressed as arbitrary units after normalization for β-actin protein expression.

### 2.4. Histopathological Examination:

The dissected mice brains were fixed in a 10% formalin solution and enclosed in paraffin. Sections of 5 µm thickness were obtained at the level of the hippocampus and cerebellum, stained with hematoxylin and eosin, and examined by a light microscope for histological examination [16].

### 2.5. Morphometric Analysis:

Morphometric measurements were performed in Cairo University-Research Park (CURP), Faculty of Agriculture, Cairo University using a Leica Qwin 500 image analyzer computer system (Leica Imaging Systems, Cambridge, England). The image analyzer consisted of a colored video camera, Panasonic wv. GP 210, colored monitor, and a hard disk of a Leica IBM personal computer connected to a BX41 Olympus microscope (Tokyo, Japan) and controlled by Lecia Qwin 500 software. Each parameter was measured by two observers blinded to the experimental group by using 10 non-overlapping readings from each animal in different groups.

The following structures were measured (Figure 1):

(1) Measurements of the molecular cell layer (M) and granular cell layer (G) thickening; at the mid sagittal section of the cerebellum; at 3 different areas; cortical thickness of the folium surface (1), cortical thickness facing the fissure (2), and cortical thickness at the fissure base (3) (Figure 1). They were performed using low magnification (10× objective, scale bar 200 µm).

(2) Morphometric analysis of the Purkinje cell count at higher magnification (40× objective, scale bar 50 µm).

(3) Morphometric analysis of pyramidal neurons counts in the CA3 region of the hippocampus (40× objective, scale bar 50 µm).

### 2.6. Transmission Electron Microscopy (TEM) Study:

Small blocks (1–2 mm thick) of the brain at the level of the hippocampus and cerebellum were fixed in 2.5% glutaraldehyde at 0.1 M sodium cacodylate buffer at 4 °C for 6 h. All procedures for the preparation and analysis of the samples were conducted according to Grimaud and Borojevic [17].

### 2.7. Statistical Analysis: 

Using SPSS 21 (IBM SPSS Statistics 21; IBM Company, New York, NY, USA), data presented as mean ± standard deviation (Mean ± SD), a comparison of quantitative variables between groups was made using variance analysis (ANOVA) with Bonferroni post-hoc test. Results were considered statistically significant for *p* ≤ 0.05 [18]. 

## 3. Results

### 3.1. Cognitive and Behavioral Tests in the Study Groups

Cognitive and behavioral tests in the ARS Group revealed an impaired exploratory behavior as indicated by the significant decrease (*p* ≤ 0.05) in the number of line crossings, the duration of central square entry, and the time spent in the central square. The ARS group also experienced increased anxiety compared to the control group as indicated by a significant increase in rearing, grooming, freezing, and stretching frequencies (*p* ≤ 0.05) compared to the control group. Ceftriaxone pretreatment significantly (*p* ≤ 0.05) improved exploratory activity and reduced anxiety relative to the ARS group (Table 2).

Working spatial memory tested by the Y-maze through the calculation of the alternating score was significantly decreased (*p* ≤ 0.05) in the ARS group compared to the control group (Figure 2).

The length of TST immobility suggesting depression in the ARS group was significantly increased (*p* ≤ 0.05) compared to the control group and the same test was significantly decreased (*p* ≤ 0.05) in the ARS + ceftriaxone group compared to the ARS group (Figure 3).

### 3.2. Results of Serum, Hippocampal, and Cerebellar Biochemical Analyses:

The biochemical analyses in blood, hippocampal, and cerebellar tissues are presented in Table 3. The serum stress markers, cortisol, IL-6, and TNF, were significantly increased (*p* ≤ 0.05) in the ARS group compared to the control group. Serum cortisol was decreased and serum IL-6 and TNF-α were significantly decreased in the ARS + ceftriaxone group compared to the ARS group. The same markers were significantly increased (*p* ≤ 0.05) in the ARS + ceftriaxone group compared to the control group.

Hippocampal BMP9, LAMP1, and GLT1 were significantly decreased (*p* ≤ 0.05) in the ARS group and the level of hippocampal HSP90a was significantly increased (*p* ≤ 0.05) in the same group compared to the control group. The levels of BMP9, LAMP1, and GLT1 were significantly (*p* ≤ 0.05) increased in the ARS + ceftriaxone group and the level of HSP90 was significantly decreased in the same group to levels that were similar to those of the control group (Table 3, Figure 4).

Cerebellar S100 B, carbonic anhydrase, and GAD were significantly (*p* ≤ 0.05) increased in the ARS group compared to the control group. The levels of cerebellar S100B and GAD were significantly (*p* ≤ 0.05) decreased in the ARS + ceftriaxone group compared to the ARS group to levels similar to those of the control group. 

Carbonic anhydrase was significantly (*p* ≤ 0.05) decreased in ARS + ceftriaxone group compared to the ARS group, although it was still significantly increased (*p* ≤ 0.05) compared to the control group. 

Cerebellar GLT1 expression was significantly (*p* ≤ 0.05) decreased in the ARS group and significantly (*p ≤ 0.05*) increased in the ceftriaxone group compared to the ARS group (Table 3, Figure 5).

### 3.3. Histological and Ultrastructural Results:

#### 3.3.1. Histopathological Examination of the H&E-Stained Sections in the CA3 Region of the Hippocampus:

The H&E-stained sections of the mouse hippocampus in the control group demonstrated the regular architecture of the CA3 region where the pyramidal cell layer neurons (P) were uniform in size and evenly arranged. Each neuron had a rounded central vesicular nucleus with a prominent nucleolus. The cytoplasm contained prominent basophilic cytoplasmic Nissel’s granules and was surrounded by thin neuropil. The molecular layer (M) contained many glial cells (G) among the neuronal processes (Figure 6A). 

The H&E-stained sections of the mouse hippocampus in the ARS group demonstrated pathologic changes in most of the nuclei of the pyramidal neurons as well as pathologic cytoplasmic changes in the form of vacuolated cytoplasm (V). Some pyramidal cells had vesicular nuclei with clogged marginated chromatin and prominent nucleoli (P) while others had pyknotic nuclei (white arrow). Few neurons had homogenous nuclei with eosinophilic cytoplasm (arrows), and others had ghost changes (G) (Figure 6B). 

The H&E-stained sections of the mouse hippocampus in the ARS group demonstrated other specimens with markedly affected pyramidal cells of the CA3 region. Most of the neurons were shrunken, with hyperchromatic nuclei (thick arrows) and vacuolated cytoplasm; others had pyknotic nuclei (thin arrows) or karyolysis (K). Areas devoid of pyramidal neurons (asterisk) were detected (Figure 6C). 

The H&E-stained sections of the mouse hippocampus in the ARS + ceftriaxone group demonstrated marked improvement of the pyramidal neurons similar to the normal architecture of the control group. They were heavily crowded with thin neuropil in between and had basophilic cytoplasm, well-formed Nissel’s granules, and vesicular nuclei (P). Few pyknotic nuclei with vacuolated cytoplasm were detected (arrows) compared to other groups (Figure 6D).

#### 3.3.2. Ultrastructural Examination in the CA3 Region of the Hippocampus

The electron micrographs of the control group demonstrated a normal ultra-structure of the pyramidal cell as the neuron had an euchromatic, central, and rounded nucleus (N) with a smooth bilaminar nuclear envelope and finely dispersed chromatin. Well-formed rough endoplasmic reticulum (R) and intact mitochondrial (M) were seen (Figure 7A). 

The ARS group demonstrated a distorted dense neuron with an irregular outline. The nucleus (N) had marginated chromatin, chromatin clumps, and was surrounded by perinuclear space (thick black arrow). The cytoplasm was shrunken and showed deposition of electron-dense bodies (e). Some mitochondria were distended with disrupted cristae (black arrow) while others were comparable to the control (white arrow). The rough endoplasmic reticulum was markedly dilated (r) (Figure 7B). 

The ARS group demonstrated markedly affected pyramidal cells. The nucleus was irregular in shape (N) and the cytoplasm contained multiple ballooned mitochondria (arrows) with disrupted cristae (Figure 7C). 

The ARS + ceftriaxone group demonstrated a marked improvement of the pyramidal cell architecture; the nucleus (N) was regular in shape and was surrounded by the bilaminar nuclear membrane. The cytoplasm revealed multiple mitochondria comparable to the control group (black arrow) while others were ruptured (white arrow) (Figure 7D).

#### 3.3.3. Histopathological Examination of H&E-Stained Sections in of the Cerebellar Cortex of Different Study Groups 

The H&E-stained sections in the mouse cerebellar tissues of the control group demonstrated the characteristic pattern of the cerebellar cortex formed of three layers: The molecular cell layer (M), Purkinje cell layer (P), and granular cell layer (G). The medulla formed of white matter fibers (W) was also seen (Figure 8A). The ARS group exhibited a marked reduction in the density of neurons in the three cortical layers (Figure 8B). The ARS + ceftriaxone group was comparable to the control group and demonstrated obvious restoration of the cellular density in the three cortical layers (Figure 8C). 

The H&E-stained sections in the mouse cerebellar tissues of the control group demonstrated the molecular cell layer (M) and showed small scattered basket cells and stellate cells. The Purkinje cell layer (P) showed large pyriform-shaped cells having vesicular open-face nuclei, eosinophilic cytoplasm, and prominent Nissl’s granules. The granular cell layer (G) showed crowded small deeply stained cells (Figure 9A). 

The ARS group demonstrated few Purkinje cells (P) with deeply stained nuclei and eosinophilic cytoplasm, surrounded by vacuolated neuropil. Other Purkinje cells were shrunken with pyknotic nuclei and vacuolated cytoplasm (arrows) (Figure 9B).

The ARS group demonstrated other specimens with marked degeneration of the Purkinje cells, exhibiting vacuolated cytoplasm (V) and pyknotic nuclei (black arrows). Basket cells and stellate cells were surrounded by perineuronal spaces (white arrows) (Figure 9C). 

The ARS + ceftriaxone group demonstrated a remarkable improvement and restoration of the normal architecture of the three cortical layers of the cerebellum; Purkinje cells were increased in number, and most of them were comparable to the control group with open face nuclei (P) while few cells had darkly stained nuclei and eosinophilic cytoplasm (arrow) (Figure 9D).

#### 3.3.4. Ultrastructural Examination of the Sections in the Cerebellar Cortex of Different Study Groups at the Purkinje Cell Layer 

The electron micrographs of the control group demonstrated Purkinje cells with an euchromatic nucleus (N) with finely dispersed chromatin and well-formed bi-laminar nuclear envelope. The cytoplasm contained scattered cell organelles. A primary dendrite (D) was also seen projecting from the cell membrane (Figure 10A). 

The ARS group demonstrated Purkinje cells with a dark electron-dense nucleus (N) and corrugated nuclear membrane. The cytoplasm contained multiple swollen mitochondria with rarified matrix and fragmented or lost cristae (M) (Figure 10B).

The ARS group demonstrated markedly affected Purkinje cells with signs of degeneration. The nucleus was shrunken, darkly electron dense, and irregular in shape with indentation of its nuclear membrane (I). Many dispersed irregularly shaped chromatin aggregates were seen throughout the nucleus. The cytoplasm contained multiple ballooned mitochondria (arrows) with disrupted cristae (Figure 10C). 

The ARS + ceftriaxone group demonstrated an improvement of the Purkinje cell architecture, with an euchromatic nucleus (N) and finely dispersed chromatin. The cytoplasm revealed multiple mitochondria comparable to the control group (M) while others were disrupted with separated outer and inner mitochondrial membranes (arrows) (Figure 10D). 

### 3.4. Morphometric Results

#### 3.4.1. Pyramidal Neurons Count in the CA3 Region of the Hippocampus

There was a significant decrease in the number of pyramidal neurons in the CA3 region of the hippocampus in mice of the ARS group compared to those of the control and ceftriaxone groups. Their number was significantly improved in the ARS + ceftriaxone group (*p* < 0.05) (Table 4).

#### 3.4.2. Thickness of Cerebellar Cortex Layers

There was a significant decrease in the thickness of the molecular and granular cell layers at three different assessed areas of the cerebellar cortex, the folium surface, facing the fissure, and at the fissure base, in mice of the ARS group compared to those of the control and ceftriaxone groups. The thicknesses of both layers (ML and GL) were significantly restored in the ARS + ceftriaxone group (*p* < 0.05) (Table 5 and Table 6).

#### 3.4.3. Purkinje Cell Count

There was a significant decrease in the number of cerebellar Purkinje cells of the ARS group compared to the control and ceftriaxone groups. The number of Purkinje cells was significantly restored in the ARS + ceftriaxone group (*p* < 0.05) (Table 7).

## 4. Discussion

Stress leads to the production of mental, emotional, and cognitive alterations [19]. Acute restraint stress (ARS) increases serum cortisol, TNF-α, and IL-6, which interacts with neurons. Stress has also been shown to decrease proliferation and neurogenesis in the hippocampus [20]. A significant part of the neuronal stress response is linked to glutamate [21]. There are connections between the stress-related regions and the cerebellum, in particular, the vermis and the midline cerebellum [22].

In the current study, ceftriaxone (CTX) (third generation cephalosporin that crosses the blood–brain barrier) demonstrated a substantial neuroprotective impact on plasma stress markers and tissue markers in the hippocampus and cerebellum. In accordance with our results, Wei et al. [23] demonstrated that CTX therapy significantly reduced the levels of proinflammatory cytokines and upregulated the expression of glutamate transporter-1. This finding indicated that ceftriaxone could be used to manage or prevent the symptoms of stress responses in the brain [21].

The pathological changes in the hippocampal and cerebellar cells in the untreated ARS group can be attributed to the high level of serum cortisol, which is involved in stress reactions. High levels of serum cortisol have previously been found to lead to multiple structural and physiological changes in the nervous system [24].

Cortisol has also been shown to influence cognition and decrease activity in the cerebellum. For example, cerebellar volume has been shown to be similarly decreased in individuals with Cushing’s disease [25]. The contribution of the cerebellum to stress-related control is linked to the existence of a high number of glucocorticoid receptors [26]. Besides poorer behavioral results on cerebellar-related tasks with high cortisol levels [27], calcium (which mediates the properties of cerebellar neurons) alterations in the cerebellum have been reported in response to acute stress exposure [28]. 

Dysmetria is thought to be related to the role of the cerebellum in sensory, cognitive, and emotional regulation [29]. Stress activates cerebellum acetylcholinesterase [30]. Cholinergic mechanisms play important functions in conscious awareness, concentration, and working memory [31], and mediate the functional activities of cerebellar neurons [32]. Additionally, acute stress promotes glutamate alterations at the cerebellar level [33] by increasing the binding of N-Methyl- d-aspartic acid (NMDA) receptors and decreasing the uptake of glutamate [34].

Cerebellar activation in response to acute stress involves the expression of the c-fos gene (also expressed in other stress-related brain areas), which was associated the action potential of the neurons [35]. Stress also increases the permeability of cerebellovascular cells [36].

Stress impairs long term potentiation (LTP) in the hippocampus. High doses of cortisol in vivo or in vitro have also been shown to impair LTP [37], indicating that cortisol is likely to mediate this stress reaction [38]. It is worth noting that the same dosage of cortisol that impairs NMDA-dependent LTP may have increased the voltage-induced calcium channel (VGCC)-dependent LTP. This source of LTP is located in the amygdala, which is thought to underly the development of fear memories [39]. 

The effect of ceftriaxone on improving behavior, cognition, and depression in the current study can be explained by the fact that ceftriaxone and corticosterone modulate specific frequency bands in opposite directions, and reveals the potential role of ceftriaxone in counteracting the effects of corticosterone [40]. Ceftriaxone might have attenuated oxidative stress markers in the hippocampus and upregulated the transporter of glutamate-1 [41], resulting in a reduction in the overload of glutamate and calcium involved in the production of reactive oxygen species (ROS) in the hippocampus [42]. This is confirmed in the current study by the effect of ceftriaxone on the expression of GLT1.

The number of crossed lines of the mice in the ceftriaxone group was significantly decreased compared to mice in the control group and those in the ARS + ceftriaxone group. This could be attributed to the upregulation of GLT1, and this is in agreement with Matos-Ocasio et al. [43].

The results of the current study showed that mice in the ARS group showed reduced hippocampal BMP9 and LAMP1 and increased hippocampal HSP90 compared to the control group. Mice in the ceftriaxone group showed increased BMP9 and LAMP1 and reduced HSP90 to levels similar to those in the control group.

Basal forebrain cholinergic neurons (BFCNs) project into the hippocampus and cerebral cortex where the activation of their neurotransmitter, acetylcholine (ACh), is essential for attention, learning, and memory processes [44]. Bone morphogenetic proteins (BMPs), particularly BMP9, are considered a cholinergic differentiation factor for BFCN [45]. BMP9 overexpression significantly decreased cell death in astrocytes [46]. BMP signaling in the hippocampus affects depressive activity [47]. 

The lysosomal membrane is implicated in the cell death pathway. Lysosomal-associated membrane protein-1 (LAMP-1), an abundant lysosomal membrane protein, produces a sugar coat or glycocalyx on the inner side of the lysosomal membrane and protects the membrane from hydrolytic enzymes and degradation [48]. LAMP1 is involved in lysosome biogenesis and autophagy [49]. Endoplasmic reticulum stress (ERS) inhibits lysosomal exocytosis and the secretion of lysosomal-associated membrane protein 1 (LAMP1) [50]. ERS is activated under stress in the hippocampus [51].

Cells react to environmental stress through the synthesis of several molecular chaperones. One of the amplest molecular chaperones is the heat shock protein 90 (Hsp90). The expression of HSPs is a sensitive marker for metabolic activation or oxidative stress [52]. Regarding the specific association of HSPs with the glucocorticoid receptor (GR), HSP90 assists GR in achieving hormone-binding conformation, where it triggers GR translocation to the nucleus and regulates the transcription of a wide range of genes. Conversely, it has been reported that HSP expression plays a protective role and offers protection to cellular stress [53]. 

In the current study, mice of the ARS group showed increased cerebellar S100 B and carbonic anhydrase relative to those in the control group. Mice in the ceftriaxone group showed normalized cerebellar S100B and decreased carbonic anhydrase compared to those in the ARS group; however, it was still upregulated compared to those in the control group. These findings may be explained by the affinity of S100 proteins to bind with calcium as a calcium-binding protein The conformational change caused by calcium binding results in the display of two adjacent hydrophobic-binding surfaces. S100 proteins also interact with calcium-dependent partner proteins, such as [54]. S100B acts as a pro-inflammatory cytokine based on its concentration. S100B secreted from astrocytes may have a trophic and toxic impact on the neurons. At nanomolar concentrations, S100B displays neurotrophic effects, leading to the promotion of neuronal survival. At micromolar concentrations, S100B induces apoptosis [55]. S100B also regulates the intracellular level of free calcium in neurons and astrocytes. S100B buffers zinc in the brain, and this is linked to a neuroprotective function through indirect effects on calcium levels and inhibition of excitotoxicity [56].

Carbon anhydrase in the nervous tissue is considered a marker for glial cells. pH has a strong modulatory effect on the function and excitability of the central nervous system. Calcium, even if catalytically inactive, may function as ‘proton-collecting antennas’, thus increasing cerebellar S100 B and carbonic anhydrase in the net transmembrane proton flux and suppressing the development of HC microdomains [57]. CA III has been shown to inhibit apoptosis in H2O2-stressed mature osteocytes [58].

Glutamic acid decarboxylase (GAD) catalyzes the formation of glutamate to *γ*-aminobutyric acid (GABA) from glutamate. GABAergic neuron dysfunction impairs the cerebellar output [59]. The neuroprotective effect of ceftriaxone may be related to restoration of the neurotransmitters’ homeostasis, including glutamate and GABA. 

## 5. Conclusions

In conclusion, the harmful effects of ARS on the hippocampus and cerebellum (both structurally and functionally) could be ameliorated by ceftriaxone, which appears to have neuroprotective properties. This deduction is supported by the improvement in the cognition and behavior of the mice, and was confirmed structurally by the preservation of the hippocampal and cerebellar architecture, as revealed by tissue markers’ expression, and microscopic and ultrastructural evaluations.

## Figures and Tables

**Figure 1 brainsci-10-00193-f001:**
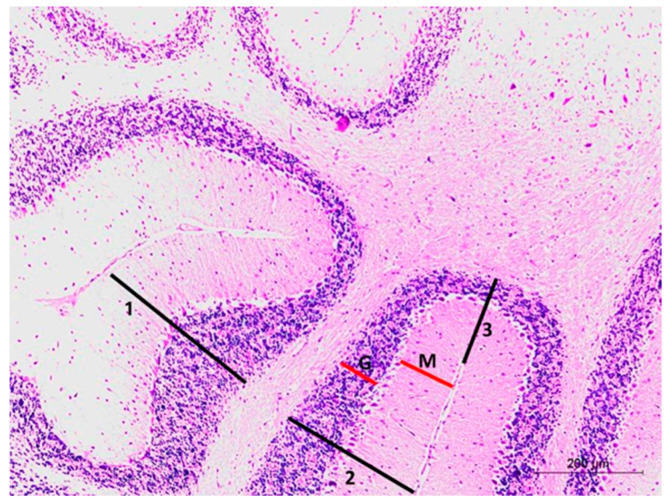
Photomicrograph of hematoxylin and eosin-stained section of the mouse cerebellar cortex at the mid-sagittal section. Measurements of the thickness of the molecular cell (M) and granular cell (G) layers were performed at three different areas: The cortical thickness of the folium surface (1), cortical thickness facing the fissure (2), and cortical thickness at the fissure base (3).

**Figure 2 brainsci-10-00193-f002:**
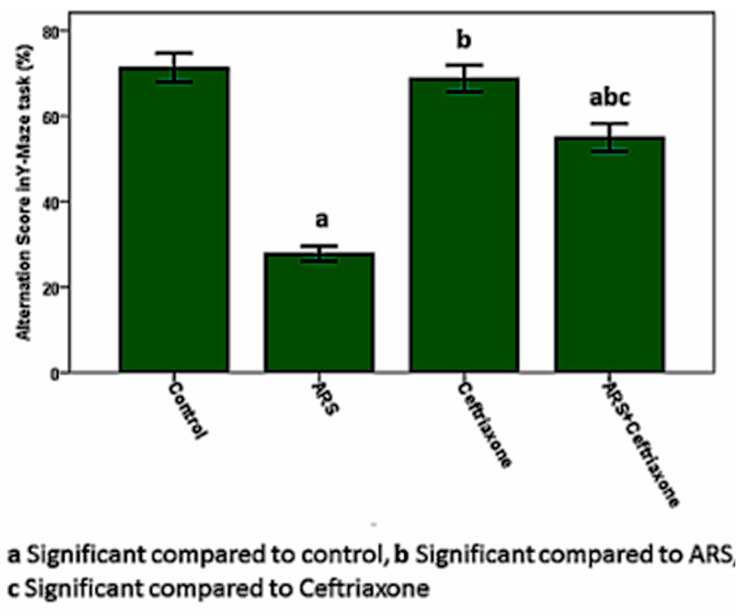
Alternation score in the Y-maze task.

**Figure 3 brainsci-10-00193-f003:**
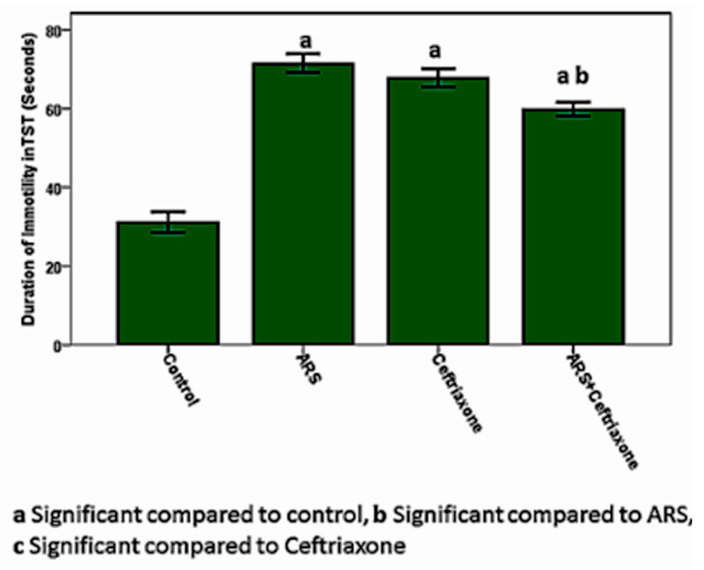
Duration of immobility in the tail suspension test.

**Figure 4 brainsci-10-00193-f004:**
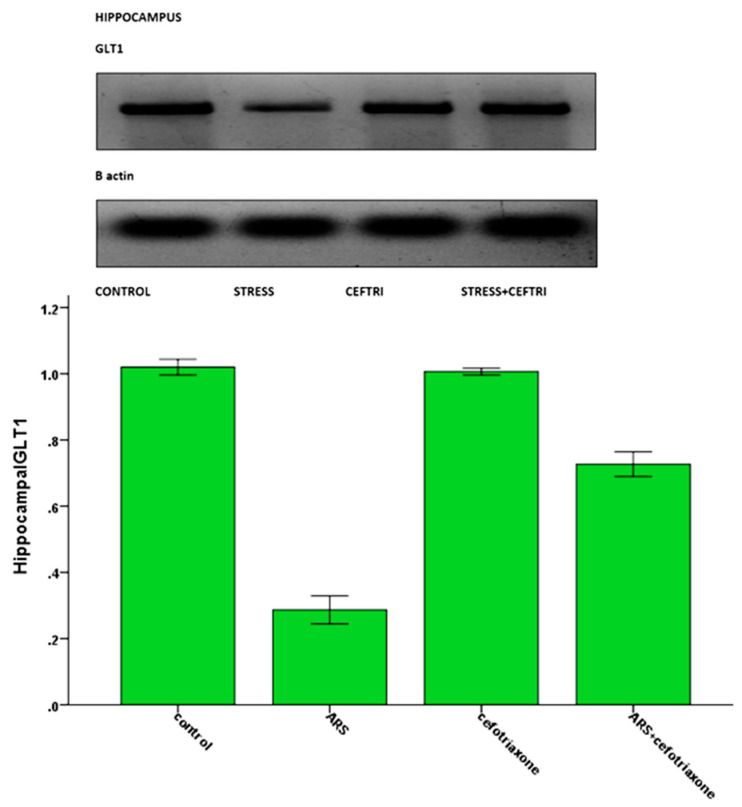
Western blot analysis of hippocampal glutamate transporter 1 expression in the studied groups.

**Figure 5 brainsci-10-00193-f005:**
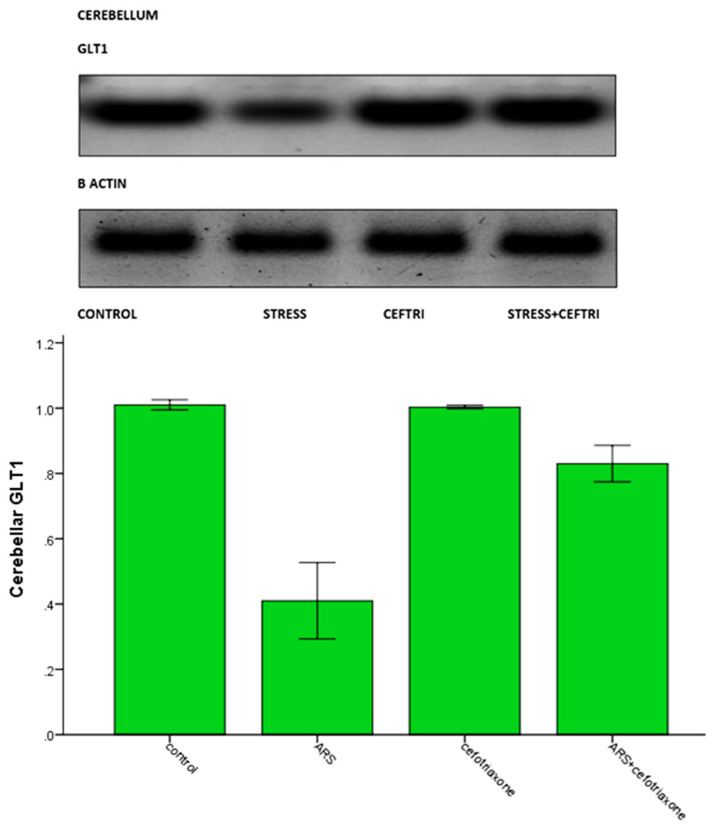
Western blot analysis of cerebellar glutamate transporter 1 expression in the studied groups.

**Figure 6 brainsci-10-00193-f006:**
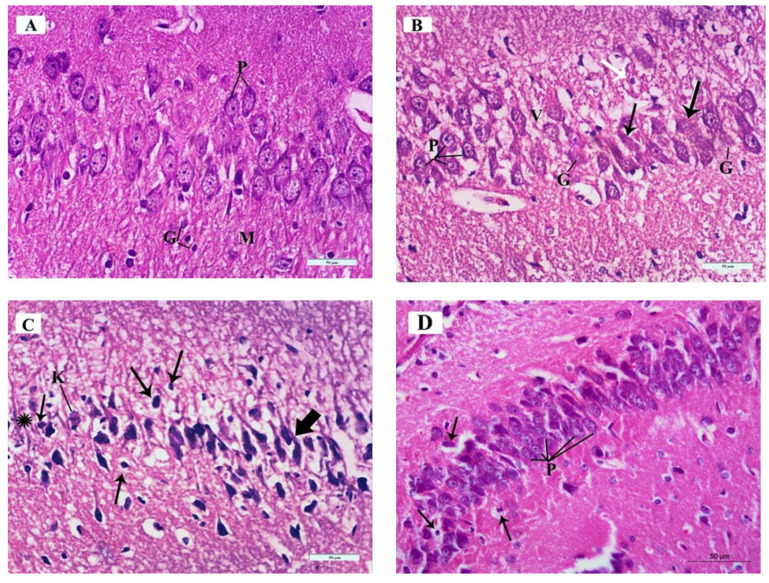
Photomicrographs of Hematoxylin and Eosin (H&E)-stained sections of the mouse hippocampus in the CA3 region of the different study groups: (**A**) Control group and ceftriaxone group; pyramidal cell-layer neurons (P) are uniform in size and evenly arranged. Each neuron has a rounded central vesicular nucleus with prominent nucleolus. The cytoplasm contains prominent basophilic cytoplasmic Nissl’s granules and is surrounded by thin neuropil. The molecular layer (M) contains many glial cells (G) among the neuronal processes. (**B**) ARS group; pleopathologic changes of most of pyramidal neurons’ nuclei as well as vacuolated cytoplasm (V). Some pyramidal cells have vesicular nuclei with clogged marginated chromatin and prominent nucleoli (P); others show pyknosis (white arrow). Few neurons show homogenous nuclei and eosinophilic cytoplasm (arrows) and others have ghost changes (G). (**C**) ARS group; most of the neurons are shrunken, with hyperchromatic nuclei (thick arrows) and vacuolated cytoplasm; others have pyknotic nuclei (thin arrows) or karyolysis (K). Areas devoid of pyramidal neurons (asterisk) are observed. (**D**) ARS + ceftriaxone group; pyramidal neurons are heavily crowded with thin neuropil in between. They have basophilic cytoplasm, well-formed Nissl’s granules, and vesicular nuclei (P). Few cells have pyknotic nuclei with vacuolated cytoplasm (arrows). (Hx. & E. × 400).

**Figure 7 brainsci-10-00193-f007:**
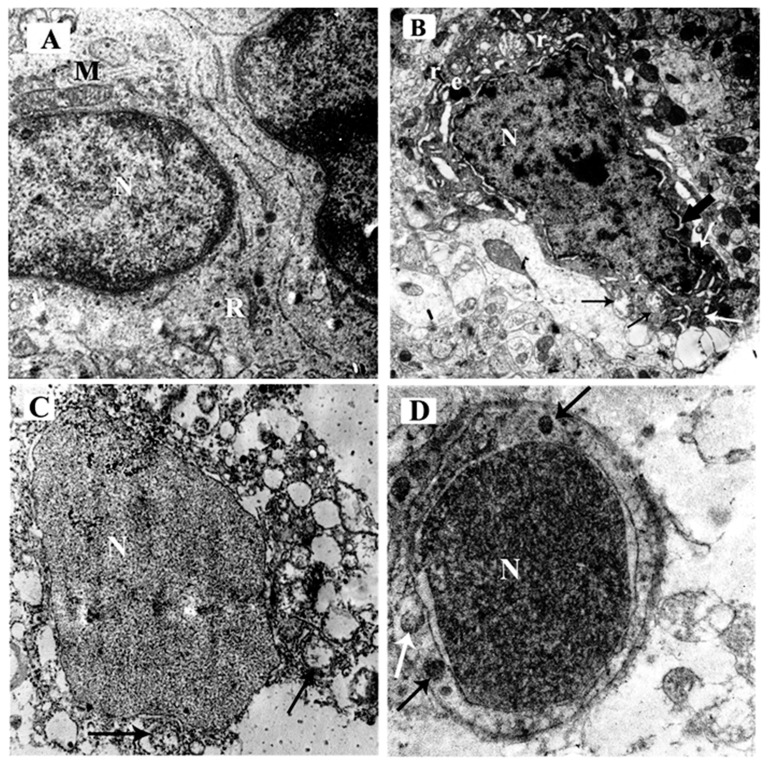
Electron micrographs of the mouse hippocampus in the CA3 region of the different study groups; (**A**) control group and ceftriaxone group; the pyramidal cell has a euochromatic, central, and rounded nucleus (N) with a smooth bi-laminar nuclear envelope and fine dispersed chromatin. Well-formed rough endoplasmic reticulum (R) and intact mitochondrial (M) are seen. (**B**) ARS group demonstrating distorted dense neuron with an irregular outline. The nucleus (N) has marginated chromatin, chromatin clumps, and is surrounded by perinuclear space (thick black arrow). The cytoplasm shows shrinkage and deposition of electron-dense bodies (e). Some mitochondria are distended with disrupted cristae (black arrow) while others are comparable to the control (white arrow). The rough endoplasmic reticulum is markedly dilated (r). (**C**) ARS group demonstrates markedly affected pyramidal cells with an irregular nucleus (N) and multiple ballooned mitochondria (arrows) with disrupted cristae. (**D**) ARS + ceftriaxone group; the pyramidal cell has a regular-shaped nucleus (N) and is surrounded by a bi-laminar nuclear membrane. The cytoplasm reveals multiple mitochondria comparable to the control (black arrow) while one is ruptured (white arrow). (× 8000).

**Figure 8 brainsci-10-00193-f008:**
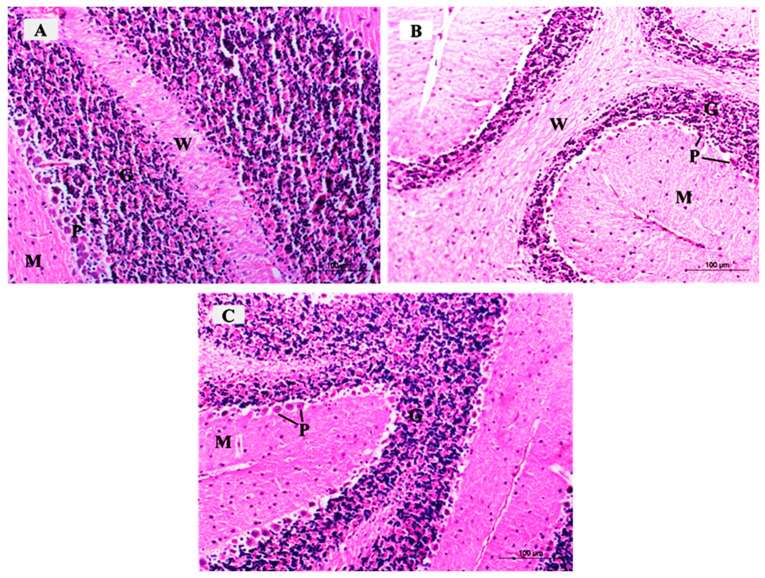
Photomicrographs of H&E-stained sections of the mouse cerebellar cortex in the different study groups; (**A)** control group and ceftriaxone group showing the molecular cell layer (M), Purkinje cell layer (P), and granular cell layer (G). The medulla formed of white matter fibers (W) is also demonstrated. (**B**) ARS group exhibits a marked reduction of the density of neurons in the three cortical layers. (**C**) ARS + ceftriaxone group demonstrates obvious restoration of the cellular density of the three cortical layers. (Hx. &E. × 200).

**Figure 9 brainsci-10-00193-f009:**
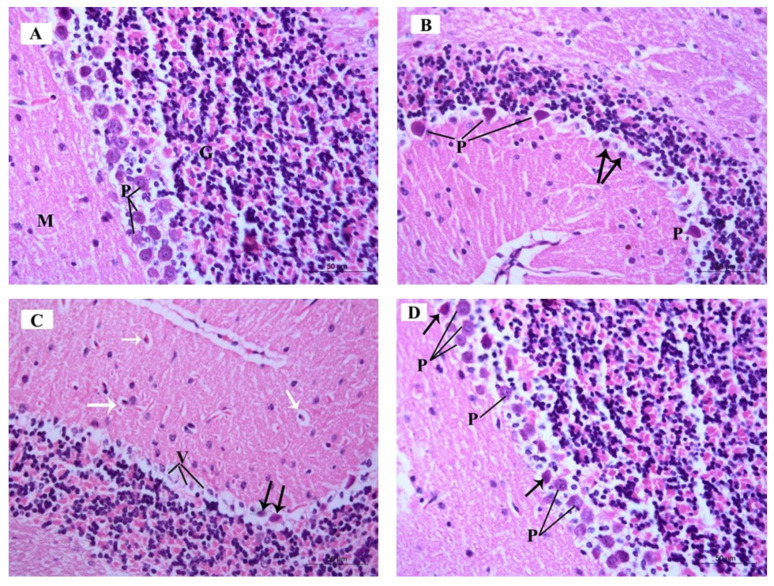
Photomicrographs of H&E-stained sections of the mouse cerebellar cortex of the different study groups: (**A**) Control group and ceftriaxone group display the molecular cell layer (M) containing small scattered basket cells and stellate cells. Purkinje cell layer (P) contains large pyriform-shaped cells having vesicular open-face nuclei and eosinophilic cytoplasm with prominent Nissl’s granules. The granular cell layer (G) contains crowded small deeply stained cells. (**B**) ARS group displays few Purkinje cells (P) with deeply stained nuclei, eosinophilic cytoplasm, and is surrounded by vacuolated neuropil. Other Purkinje cells are shrunken with pyknotic nuclei and vacuolated cytoplasm (arrows). (**C**) ARS group shows Purkinje cells having vacuolated cytoplasm (V) and pyknotic nuclei (black arrows). Basket cells and stellate cells are surrounded with perineuronal spaces (white arrows). (**D**) ARS + ceftriaxone group. Purkinje cells are increased in number; most of them display open-face nuclei (P) while few of them have darkly stained nuclei and eosinophilic cytoplasm (arrow). (Hx. &E. × 400).

**Figure 10 brainsci-10-00193-f010:**
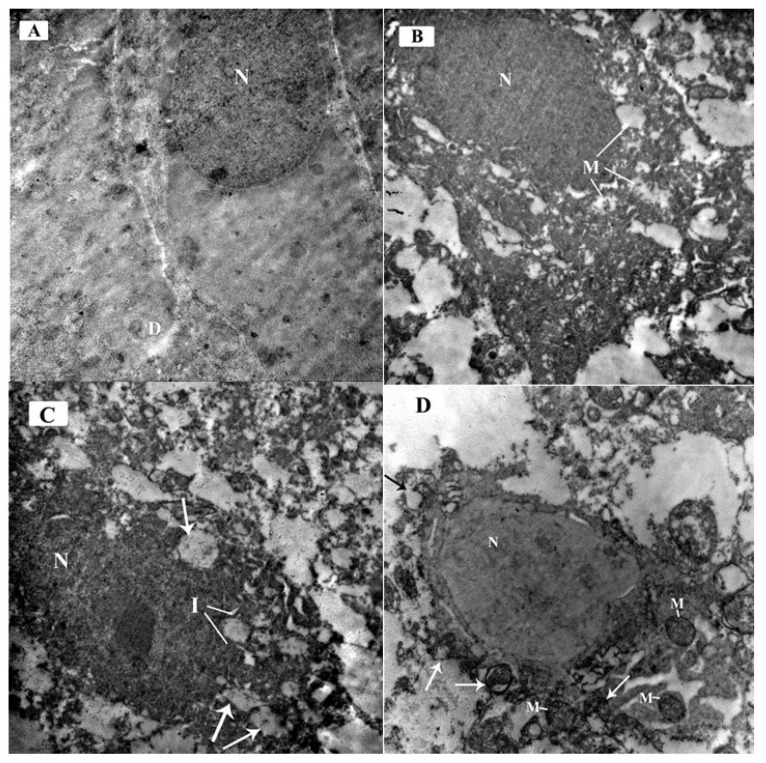
Electron micrographs of the mouse cerebellar cortex in different study groups at the Purkinje cell layer: (**A**) Control group and ceftriaxone group; Purkinje cell has a euochromatic nucleus (N) with fine dispersed chromatin and a well-formed bi-laminar nuclear envelope. The cytoplasm contains scattered cell organelles. A primary dendrite (D) is also seen projecting from the cell membrane. (**B**) ARS group; Purkinje cell have a dark electron-dense nucleus (N) with an irregular nuclear membrane. The cytoplasm contains many swollen mitochondria showing rarified matrix with fragmented or lost cristae (M). (**C**) ARS group displays markedly affected Purkinje cells. The nucleus is shrunken, darkly electron-dense, and irregular in shape with indentation of its nuclear membrane (I). Many dispersed irregularly shaped chromatin aggregates are demonstrated throughout the nucleus. The cell has irregular outline with shrunken cytoplasm containing multiple ballooned mitochondria (arrows) with disrupted cristae. (**D**) ARS + ceftriaxone group; Purkinje cell exhibits a euochromatic nucleus (N) and fine dispersed chromatin. The cytoplasm reveals multiple mitochondria comparable to the control (M) while others are disrupted with separated outer and inner mitochondrial membranes (arrows). (× 6000).

**Table 1 brainsci-10-00193-t001:** The primer sequence of the studied gene.

	Primer Sequence
**BMP-9**	Forward: 5’-- TTCAGGATGAGGGCTGGGAG -3’ Reverse: 5’- GGATGTCTTCACAAGCACGGTC 3’
**Lamp1**	Forward: 5’- GTCCTCATCGTCCTCATTGC -3 Reverse: 5’- CTGATAGCCGGCGTGACT-3
**S100**	Forward: 5’ TGCTGTGGTTGGCATTTT TC 3’ Reverse: 5’ AGGCTGCGCAGCTTGGCCAT 3’
**CAII**	Forward: 5’ TGCCCTCAGTTTGTGCAGAATA 3’ Reverse: 5’ CCAACGCAAGGAACTCTTCGA3’
**HSP90**	Forward:5’--GGTCATCTTGCTGTACGAAA -3 Reverse: 5′- GGTGGCATTTCTTCAGTTAC-3
**GLT- 1**	Forward: 5-GAAAAAACCCATTCTCCTTTTT-3 Reverse:5- CCGACTGGGAGGACGAATC-3
**GAD**	Forward: 5-GAAAAAACCCATTCTCCTTTTT-3 Reverse: 5- CCGACTGGGAGGACGAATC-3
**Beta actin**	Forward: 5’--GGTCGGTGTGAACGGATTTGG -3 Reverse: 5′- ATGTAGGCCATGAGGTCCACC-3

**Table 2 brainsci-10-00193-t002:** Performance in open field and Y-maze tests in the studied groups.

Test		Control Group	ARS Group	Ceftriaxone Group	ARS + Ceftriaxone Group
**Open field**	**Number of crossed lines**	81.17 ± 2.483	19.17 ± 2.137 ^a^	70.50 ± 5.612 ^a,b^	82.83 ± 3.971 ^b,c^
	**Central Square entry Frequency**	8.50 ± 1.049	1.33 ± 0.816 ^a^	7.83 ± 1.169 ^b^	7.17 ± 1.169 ^b^
	**Central Square Duration (Seconds)**	20.17 ± 1.941	3.33 ± 1.211 ^a^	16.83 ± 1.472 ^a,b^	12.67 ± 1.366 ^a,b,c^
	**Rearing Frequency**	5.17 ± 0.983	14.17 ± 1.722 ^a^	4.33 ± 1.751 ^b^	10.83 ± 1.472 ^a,b^
	**Grooming Frequency**	2.33 ± 0.816	9.00 ± 1.095 ^a^	2.00 ± 1.095 ^b^	4.33 ± 0.816 ^a,b,c^
	**Freezing Frequency**	2.00 ± 0.632	8.83 ± 1.169 ^a^	2.50 ± 0.548 ^b^	5.00 ± 1.095 ^a,b,c^
**Y-Maze**	total number of arm entries	26 ± 2.832	14 ± 2.22 ^a^	24 ± 1.89 ^b^	19 ± 2.53 ^a,b,c^
	**Alternation score (%)**	75 ± 3.410	28 ± 1.260 ^a^	73 ± 3.002 ^b^	56 ± 3.1 ^a,b,c^

^a^: Significant compared to control, ^b^: Significant compared to ARS, ^c^: Significant compared to Ceftriaxone.

**Table 3 brainsci-10-00193-t003:** Biochemical analyses of serum hippocampal and cerebellar tissues in the studied groups.

	Control Group	ARS Group	Ceftriaxone Group	ARS + Ceftriaxone Group
**Serum TNF (pg/mL)**	14.433 ± 0.4131	112.367 ± 0.16.0814 ^a^	16.367 ± 0.2.9588 ^b^	52.700 ± 0.22.0782 ^a,b,c^
**Serum IL6 (pg/mL)**	21.400 ± 0.2280	95.267 ± 20.5213 ^a^	28.133 ± 2.0675 ^b^	60.70 ± 4.3964 ^a,b,c^
**Serum cortisol (ug/mL)**	4.1850 ± 0.22941	9.810 ± 0.80811 ^a^	3.6867 ± 0.39450	5.470 ± 0.43626 ^b,c^
**Hippocampal BMP9 (pg/mg protein)**	0.9833 ± 0.07528	0.483 ± 0.26838 ^a^	1.016 ± 0.01862 ^b^	0.8733 ± 0.09812 ^b,c^
**Hippocampal HSP90 (pg/mg protein)**	1.100 ± 0.12649	5.0067 ± 0.98392 ^a^	1.010 ± 0.00894 ^b^	1.800 ± 0.26833 ^b^
**Hippocampal LAMP1 (ng/mg protein)**	1.050 ± 0.19748	0.460 ± 0.26668 ^a^	1.0167 ± 0.02582 ^b^	0.880 ± 0.16757 ^b^
**Hippocampal GLT1(PCR)**	1.0067 ± 0.01	0.1867 ± 0.02 ^a^	1.0367 ± 0.03 ^b^	0.6233 ± 0.03 ^a^^,^^b^^,^^c^
**Hippocampal GLT1(WB)**	1.02 ± 0.023	0.2867 ± 0.042 ^a^	1.0067 ± 0.0103 ^b^	0.7267 ± 0.037 ^a,b,c^
**Cerebellar S100 B (pg/mg protein)**	6.133 ± 0.2251	16.133 ± 5.6811 ^a^	6.133 ± 0.7711 ^b^	7.633 ± 1.1911 ^b^
**Cerebellar Carbonic Anhydrase (ng/mg protein)**	2.3167 ± 0.37639	8.700 ± 1.35351 ^a^	2.6033 ± 0.32745 ^b^	5.300 ± 0.77974 ^a,b,c^
**Cerebellar GLT1 (PCR)**	1.0033 ± 0.005	0.3267 ± 0.0981 ^a^	1.0067 ± 0.0103 ^b^	0.7333 ± 0.0403 ^a,b,c^
**Cerebellar GLT1 (WB)**	1.010 ± 0.0154	0.4100 ± 0.1169 ^a^	1.0033± 0.0051 ^b^	0.830 ± 0.0558 ^a,b,c^
**Cerebellar GAD (PCR)**	1.013 ± 0.0206	4.6000 ± 0.3224 ^a^	1.0133 ± 0.0206 ^b^	2.6367 ± 0.31879 ^a,b,c^

^a^: Significant compared to control, ^b^: Significant compared to ARS, ^c^: Significant compared to ceftriaxone. TNF = tumor necrotic factor, IL6 = interleukin-6, BMP9 = Bone morphogenetic protein 9, HSP90 = Heat shock protein 90, LAMP1 = Lysosomal-associated membrane protein 1, GLT1 = Glutamate transporter 1, GAD = glutamic acid decarboxylase.

**Table 4 brainsci-10-00193-t004:** Morphometric analysis of the pyramidal neuron count in the CA3 region of the hippocampus.

Groups	Mean ± SD	Pairwise Comparisons with Other Groups	
**Control**	41.83 ± 6.21	ARS	*p* < 0.0015
		Cef.	*p* < 0.7194
		ARS+Cef	*p* < 0.1359
ARS	26.00 ± 6.42	Control	*p* < 0.0015
		Cef.	*p* < 0.0027
		ARS+Cef	*p* < 0.0107
**Ceftriaxone**	40.50 ± 6.28	Control	*p* < 0.7194
		ARS	*p* < 0.0027
		ARS+Cef	*p* < 0.2549
ARS + Ceftriaxone	36.50 ± 5.13	Control	*p* < 0.1359
		ARS	*p* < 0.0107
		Cef.	*p* < 0.2549

**Table 5 brainsci-10-00193-t005:** Morphometric analysis of the molecular cell layer thickness at three different areas of the cerebellar folia.

Groups	Mean ± SD	Pairwise Comparisons with Other Groups		Mean ± SD	Pairwise Comparisons with Other Groups		Mean ± SD	Pairwise Comparisons with Other Groups	
	At the Folium Surface			Facing the Fissure			At the Fissure Base		
Control	83.15± 5.77	ARS	*p* < 0.0020	125.0 ± 5.49	ARS	*p* < 0.0043	131.18 ± 5.99	ARS	*p* < 0.0005
		Cef.	*p* < 0.9679		Cef.	*p* < 0.6151		Cef.	*p* < 0.8693
		ARS+Cef	*p* < 0.4147		ARS+Cef	*p* < 0.2207		ARS+Cef	*p* < 0.1152
ARS	98.74± 7.17	Control	*p* < 0.0020	138.28 ± 6.90	Control	*p* < 0.0043	147.88 ± 5.55	Control	*p* < 0.0005
		Cef.	*p* < 0.0019		Cef.	*p* < 0.0110		Cef.	*p* < 0.0003
		ARS+Cef	*p* < 0.0116		ARS+Cef	*p* < 0.0298		ARS+Cef	*p* < 0.0060
Cef.	83.28± 5.57	Control	*p* < 0.9679	126.76 ± 5.87	Control	*p* < 0.6151	130.62 ± 5.42	Control	*p* < 0.8693
		ARS	*p* < 0.0019		ARS	*p* < 0.0110		ARS	*p* < 0.0003
		ARS+Cef	*p* < 0.4280		ARS+Cef	*p* < 0.4755		ARS+Cef	*p* < 0.1160
ARS + Ceftriaxone	86.26± 6.85	Control	*p* < 0.4147	129.18 ± 5.46	Control	*p* < 0.2207	136.88 ± 5.44	Control	*p* < 0.1152
		ARS	*p* < 0.0116		ARS	*p* < 0.0298		ARS	*p* < 0.0060
		Ceftriaxone	*p* < 0.4280		Ceftriaxone	*p* < 0.4755		Ceftriaxone	*p* < 0.1160

**Table 6 brainsci-10-00193-t006:** Morphometric analysis of the granular cell layer thickness at three different areas of the cerebellar folia.

Groups	Mean ± SD	Pairwise Comparisons with Other Groups		Mean ± SD	Pairwise Comparisons with Other Groups		Mean ± SD	Pairwise Comparisons with Other Groups	
	At the Folium Surface			Facing the Fissure			At the Fissure Base		
Control	122.67± 4.73	ARS	*p* < 0.0002	99.20± 5.87	ARS	*p* < 0.0004	73.20± 5.93	ARS	*p* < 0.0005
		Cef.	*p* < 0.8941		Cef.	*p* < 0.7739		Cef.	*p* < 0.9376
		ARS+Cef	*p* < 0.3302		ARS+Cef	*p* < 0.2117		ARS+Cef	*p* < 0.2181
ARS	104.21± 6.28	Control	*p* < 0.0002	82.20± 5.37	Control	*p* < 0.0004	54.18± 7.14	Control	*p* < 0.0005
		Cef.	*p* < 0.0001		Cef.	*p* < 0.0001		Cef.	*p* < 0.0002
		ARS+Cef	*p* < 0.0009		ARS+Cef	*p* < 0.0044		ARS+Cef	*p* < 0.0046
Cef.	123.01± 3.85	Control	*p* < 0.8941	98.36± 3.68	Control	*p* < 0.7739	73.44± 4.29	Control	*p* < 0.9376
		ARS	*p* < 0.0001		ARS	*p* < 0.0001		ARS	*p* < 0.0002
		ARS+Cef	*p* < 0.2434		ARS+Cef	*p* < 0.2242		ARS+Cef	*p* < 0.1499
ARS+ Cef	119.72± 5.25	Control	*p* < 0.3302	94.52± 6.27	Control	*p* < 0.2117	68.48± 6.50	Control	*p* < 0.2181
		ARS	*p* < 0.0009		ARS	*p* < 0.0044		ARS	*p* < 0.0046
		Cef.	*p* < 0.2434		Cef.	*p* < 0.2242		Cef.	*p* < 0.1499

**Table 7 brainsci-10-00193-t007:** Morphometric analysis of the Purkinje cells count.

Groups	Mean ± SD	Pairwise Comparisons with Other Groups	
Control	21.33 ± 2.80	ARS	*p* < 0.0001
		Cef.	*p* < 0.4969
		ARS+Cef	*p* < 0.0658
ARS	10.83 ± 3.19	Control	*p* < 0.0001
		Cef.	*p* < 0.0004
		ARS+Cef	*p* < 0.0031
Ceftriaxone	20.17 ± 2.93	Control	*p* < 0.4969
		ARS	*p* < 0.0004
		ARS+Cef	*p* < 0.2069
ARS + Ceftriaxone	17.83 ± 3.06	Control	*p* < 0.0658
		ARS	*p* < 0.0031
		Cef.	*p* < 0.2069

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
