# Peer review of "Hippocampal and Cerebellar Changes in Acute Restraint Stress and the Impact of Pretreatment with Ceftriaxone"

_brainsci, 2020, doi:10.3390/brainsci10040193_

Round 1

Reviewer 1 Report

The Authors performed new experiments addressing most of my requests.

My only new request is related to the quality of the representative GLT-1 immunoblots, which are overexposed. The Authors should be use a blot with a lower exposure time (both GLT-1 and actin for the hippocampus and actin for the cerebellum). If not available, they can repeat the blots loading a lower amount of proteins or reducing the exposure time.

Author Response

Response: We replaced it with a new one with lower exposure.

Reviewer 2 Report

Although most concerns have been adequately addressed, I would appreciate it if the authors would also respond to all comments provided by English editing services.

Author Response

Response: manuscript revised an edited by an expert American with phD

This manuscript is a resubmission of an earlier submission. The following is a list of the peer review reports and author responses from that submission.

Round 1

Reviewer 1 Report

This manuscript reports the protective action of ceftriaxone against behavioral abnormalities and structural and biochemical changes induced by acute restraint stress in mice. On one side, the multitasking approach of the study deserves some credit. However, there are several flaws that mitigate my enthusiasm for this manuscript. I have the following requests that should be addressed before the manuscript is considered for further evaluation.

It is hard to believe that 2.5 hours of immobility cause such dramatic morphological changes in the hippocampus and cerebellum. Unbiased assessments of cell morphology must be performed, particularly in the cerebellum where the thickness of the cortical layers is not uniform in the various subregions. It is important to quantify neuronal damage or loss in multiple sections by a blind observer (if stereological cell counting cannot be performed). The choice of the biochemical markers is justified in the Discussion, but is not entirely appropriate. One mechanism that might be relevant to the protective action of ceftriaxone is the induction of the glutamate transporter GLT-1. This should be measured by qPCR and/or WB. The catalytic subunit (xCT) of the cystine-glutamate antiporter should also be measured. Quantification of GAD in the cerebellum will be a reliable biochemical correlate of neuronal loss in the Purkinje cell and molecular layers. There are many typos that should be corrected. Some sentences should also be edited. For example, it is stated in the last sentence of the Abstract that the effect of ARS are accentuated by ceftroaxone (??), which is in contrast with all data shown in the manuscript. An extensive editing of the manuscript is required.  

Reviewer 2 Report

This manuscript entitled “Hippocampal and cerebellar changes in acute restraint stress and the impact of pretreatment with ceftriaxone” by Amin et al. reports that the effects of ceftriaxone on acute stress. Although the potential of neuroprotection with ceftriaxone to treat stress-related disorders is very interesting, the provided data are not thought to fully support the author’s interpretation of results. I think additional data are needed and some modifications in some of the conclusions are warranted.

The authors give only a description of the keyword in the Introduction section, lacks explanation about the Table 2, Figure 1 and 2 in the Results section, and do not fully include the authors’ interpretations and opinions, explaining the effects of provided findings, and making suggestions and predictions for future research and so on in the Discussion section. Figure legends are also largely unexplained about the Figures. Besides, there are multiple grammatical and spelling errors and typos throughout, and so the manuscript will require careful editing for English and spelling. Please add information of mice used in the Materials and Methods section. Were Control and ARS groups injected with the vehicle that dissolved ceftriaxone? In the Open field test (Table 2), why is the number of crossed lines of ceftriaxone-pretreated mice significantly less than that of control mice and ARS + ceftriaxone group? In the Y maze test (Figure 1), the alternation ratio in arm choices of ARS + ceftriaxone mice is significantly higher than that of ARS, but is the value above chance level (e.g. 50%)? I think ARS + ceftriaxone mice do not exhibit a significant alternation ratio. Also, the alternation ratio of ARS is “much less” than the chance level. This result is different from previous studies (e.g. Ohgidani et al., Brain Behav Immun., 2016; Woo et al., Mol Brain, 2018). Any explanation about these discrepancies? Also, please add data for the total number of arm entries of all groups in Y maze tests. In the tail suspension test (Figure 2), why is the duration of immobility of ceftriaxone-treated mice is significantly higher than that of control? Nevertheless, the duration of immobility of ARS + ceftriaxone mice is significantly lower than that of ARS mice. When did the authors conduct the behavioral test and collect the sample for biochemical measurement and histological analysis after the stress session? In Table 3, RT-PCR does not quantify the protein level of BMP9, HSP, LAMP1, S100β, and Carbonic anhydrase, but messenger RNA. In histological analysis (Figure 3-6), please check the total number of cells and whether the apoptosis is evoked, and also please provide not only qualitative data but also quantitative data. Besides, please add the data for ceftriaxone-treated mice. It would be extremely informative if the authors provide data on the effects of CLT-1 of ceftriaxone on acute stress (Ref. Romos et al., Neuroscience, 2010; Hu et al., J Neurochem., 2015; Krzyżanowska et al., PLoS One, 2017; Lacrosse et al., J Neurosci., 2017).